# Magnetic Field-Assisted Orientation and Positioning of Magnetite for Flexible and Electrically Conductive Sensors

**DOI:** 10.3390/mi16010068

**Published:** 2025-01-08

**Authors:** David Seixas Esteves, Amanda Melo, Sónia Alves, Nelson Durães, Maria C. Paiva, Elsa W. Sequeiros

**Affiliations:** 1Department of Mechanical Engineering, Faculty of Engineering, University of Porto, 4200-465 Porto, Portugal; ews@fe.up.pt; 2CeNTI, Centre for Nanotechnology and Advanced Materials, 4760-034 Vila Nova de Famalicão, Portugal; amelo@centi.pt (A.M.); salves@centi.pt (S.A.); nduraes@centi.pt (N.D.); 3Department of Polymer Engineering, IPC—Institute for Polymers and Composites, University of Minho, 4800-058 Guimarães, Portugal; 4LAETA/INEGI—Institute of Science and Innovation in Mechanical and Industrial Engineering, 4200-465 Porto, Portugal

**Keywords:** magnetite, MWCNT, flexible electronics, conductive composites

## Abstract

Magnetic field-assisted control of magnetite location is a promising strategy for developing flexible, electrically conductive sensors with enhanced performance and adjustable properties. This study investigates the effect of static magnetic fields applied on thermoplastic elastomer (TPE) composites with magnetite and multi-walled carbon nanotubes (MWCNT). The composites were prepared by compression moulding and the magnetic field was applied on the mould cavity during processing. Composites were prepared with a range of concentrations of magnetite (1, 3, and 6 wt.%) and MWCNT (1 and 3 wt.%). The effect of particle concentration on composite viscosity was investigated. Rheological analysis showed that MWCNTs significantly increased the composite viscosity while magnetite had minimal impact, ensuring stable processing and facilitating particle orientation under a static magnetic field. Particle orientation and electrical conductivity were evaluated for the composites prepared with different particle concentrations under different processing temperatures. Magnetic field application at 190 °C enhanced magnetite/MWCNT interactions, substantially reducing electrical resistivity while preserving thermal stability. The composites showed no degradation at 220 °C and above, demonstrating suitability for high-temperature applications requiring thermal resilience. Furthermore, magnetite’s magnetic response facilitated precise sensor positioning and strong adhesion to polyimide substrates at 220 °C. These findings demonstrate a scalable and adaptable approach for enhancing sensor performance and positioning, with broad potential in flexible electronics.

## 1. Introduction

The field of composite materials has advanced significantly in recent decades, driven by research and innovations in manufacturing and advanced materials design [1]. Among these innovations, incorporating magnetic and conductive fillers into a polymer matrix is a promising approach for enhancing and diversifying the functional capabilities of flexible composite materials. The ability to control the orientation of these fillers within the matrix can substantially improve mechanical strength [2,3], thermal stability [4], and electrical conductivity [5,6]. The use of thermoplastic matrices in the development of flexible and conductive composites has demonstrated key advantages relative to other matrix materials, offering superior processability, recyclability, and adaptability [7]. Thermoplastic elastomers (TPE) combine flexibility with ease of moulding and reshaping, making them particularly well-suited for applications demanding both adaptability and sustainability. This contrasts with thermoset polymers, which, despite their stiffness, lack reusability after curing, limiting their versatility and reducing their alignment with modern sustainability goals [8].

Therefore, this study introduces a novel methodology for controlling filler orientation within TPE polymer matrices by employing magnetically induced particle alignment. Specifically, magnetite (Fe_3_O_4_) particles were chosen for their inherent magnetic responsiveness, which facilitates their precise orientation within the polymer matrix when subjected to an external magnetic field [9,10]. This targeted orientation optimises the distribution of fillers, which could improve the overall performance of the composite. While magnetite alone provides moderate electrical conductivity in combination with magnetic properties, the level of conductivity required for advanced flexible electronics often necessitates a hybrid material approach [9,10]. To address this, magnetite may be combined with multi-walled carbon nanotubes (MWCNTs), which possess exceptionally low resistivity [11,12]. Integrating MWCNTs can significantly reduce the composite electrical resistance, even at minimal concentrations, while maintaining magnetic responsiveness [9,10,11,12].

Several techniques are available to enhance composite electrical conductivity through particle orientation, each with specific mechanisms that influence particle alignment and significantly impact the material’s properties [3,13,14,15,16,17,18,19,20,21,22]. Among these, magnetic field orientation presents unique advantages, especially for magnetic or magnetically responsive particles. By leveraging the particles’ magnetic susceptibility, a magnetic field can be applied during composite fabrication to align particles along the field lines, forming an organised internal structure. Unlike mechanical or electric field methods, magnetic orientation is non-intrusive and can be applied uniformly across the material, enabling fine control over particle alignment and distribution [23,24,25,26,27]. In addition to potential enhancements associated with controlled particle orientation, the multifunctional characteristics of magnetite have drawn substantial interest in a wide range of electronic applications [28]. In a sense, magnetite nanoparticles enhance systems for detecting magnetic fields and other critical targets, making them highly valuable in medical diagnostics [28]. In the energy sector, magnetite’s high theoretical capacity makes it a promising candidate for use as an anode material in lithium-ion batteries. Additionally, it exhibits intriguing properties for wireless charging technologies [29,30,31]. Moreover, magnetite’s magnetic loss properties render it highly effective for electromagnetic interference (EMI) shielding, mainly when employed in combination with carbon-based materials [32,33].

In the present work, TPE hybrid composites incorporating magnetite and MWCNT were produced by melt processing, a method that is highly scalable and environmentally friendly. Using low filler contents, the composites were moulded under the effect of magnetic fields to induce particle/nanoparticle orientation and enhance the composite response under different stimuli. Therefore, the work presented contributes to the development of multifunctional composites with potential applications in several fields, including sensors, actuators, electromagnetic shielding, soft robotics and flexible electronics [34,35,36,37,38].

## 2. Materials and Methods

This study examined TPE composites reinforced with magnetite and MWCNT and studied the influence of these fillers on the composite’s morphology, rheology, and electrical properties. The workflow diagram in Figure 1 illustrates the sequential stages of composite preparation, characterisation, and testing employed in this study. The first stage was the composite materials preparation where magnetite, multi-walled carbon nanotubes (MWCNTs), and thermoplastic elastomer (TPE) are combined by melt mixing using twin-screw extrusion. From these composites, samples were moulded with and without the application of magnetic fields to investigate its effect on particle distribution. The next stage consisted of the composites’ characterisation using different analytical techniques to assess the morphology, structure and properties of the composites. Morphological analysis was conducted by Scanning Electron Microscopy (SEM) and Micro Computed Tomography (Micro-CT) to visualise particle distribution and orientation within the composite. Rheological analysis of the composite melt assessed the viscosity, loss and storage moduli, providing insight into the effect of the addition of magnetite and MWCNTs on the viscoelastic behaviour of the TPE composite. Thermogravimetric analysis (TGA) characterised the thermal stability of the composites, defining their maximum processing temperature. In the final stage, electrical and functional testing was conducted on the final composites to compare the electrical resistivity with and without magnetic field application and assess the impact of magnetic alignment on conductivity. Finally, a laser cutting technique was used to create sensor patterns in the composite films. These patterned films were then bonded to flexible substrates with magnetic assistance. The stability and performance of the sensors were evaluated through bending cycles, simulating repeated mechanical stress to assess adhesion durability.

### 2.1. Materials

MWCNTs used in this study were obtained from Nanocyl, the NC 7000 grade (Sambreville, Belgium). The TPE employed was Engage 8401, a high-performance elastomer supplied by Dow Chemical Company (Midland, MI, USA). Rectangular Samarium Cobalt (SmCo) permanent magnets with dimensions of 25 × 9.5 × 12 mm were acquired from Superimanes S.L. (Madrid, Spain). Synthetic magnetite was purchased from Inoxia Ltd. (Alton, Hampshire, UK), providing magnetic particles for composite production.

### 2.2. Composite Production

Composite production by extrusion: For the sample production, five distinct composites were produced using a twin-screw extruder, incorporating 1%, 3%, and 6% by weight (wt.%) of magnetite and 1% and 3% by weight (wt.%) of MWCNT. Subsequently, hybrid composites were prepared as follows: 6% of magnetite was added to 500 g of the 1% or 3% MWCNT composites, to fabricate composites containing 1% or 3% wt.% MWCNT with 6 wt.% magnetite each. The magnetite powder was added to the TPE/MWCNT composite using a gravimetric dosing system by Movacolor B.V. (Sneek, The Netherlands), calibrated beforehand for accuracy. The filaments were produced using a co-rotating twin-screw extruder with a length-to-diameter (L/D) ratio of 25 from Rondol Technology Ltd. (Staffordshire, UK). The extrusion process was conducted under specific conditions: the temperature along the extruder was set at 140 °C for the first and last zones and 145 °C for zones 2 through 4. The drive torque was maintained at 30%, and the screw speed was set to 150 rpm.

Composite sample preparation by compression moulding: To ensure accurate process control, the mould cavity temperature was initially monitored due to its importance in the thermal dynamics and resulting properties of the manufactured material. Temperature variations between the set values on processing equipment and real mould cavity temperatures can impact material properties. A Type J thermocouple connected to a UNI-T UT320 A (Dongguan City, Guangdong Province, China) thermometer was used to monitor cavity temperature, with data recorded every 30 s for a 20 min duration at each set temperature. Results showed significant discrepancies between the set and real cavity temperatures, with measured temperatures consistently lower within the cavity. For example, a set temperature of 160 °C resulted in a cavity temperature of approximately 130 °C, while set temperature of 190 °C and 220 °C yielded cavity temperatures of approximately 160 °C and 190 °C, respectively, after 20 min.

The samples were prepared from the extruded composites using a stainless-steel mould with a thickness of 3 mm. Each mould contained two 40 × 20 × 3 mm cavities, one designated for samples with permanent magnets and another as a control. Samples were processed using compression moulding (Galaxy XL5880, Acosgraf, Canelas, Portugal) at specified temperatures and durations, as detailed in Table 1 and Appendix A, applying a constant pressure of 2 bar to protect the integrity of the permanent magnets.

Magnetic field application on the mould cavity: The first two SmCo permanent magnets (25 × 9.5 × 12 mm each), were aligned in a north–south (NS) configuration and securely attached. Two additional magnets, arranged in a south–north (SN) configuration, were positioned parallel to the first pair, spaced 3 mm apart, as shown in Figure 2. All four magnets were firmly fixed to a 5 mm thick iron base to ensure stability during the composite fabrication process. Precise alignment of the magnets in the NSNS configuration was essential for achieving a continuous and uniform particle alignment within the composite material [6].

The magnetic properties of the SmCo magnets were characterised using a 3D magnetic field mapper (M3D-2A-Port, Senis, Canton Zug, Switzerland) over a mapped area of 75 × 35 mm. Measurements were taken at two key distances: 0.2 mm and 3 mm from the magnet surface. At 0.2 mm, the magnetic flux density peaked at 400 mT, while at 3 mm, it decreased to approximately 200 mT, demonstrating the expected attenuation with distance. These measurements confirmed the consistency and reliability of the magnetic field, which is essential for effective particle alignment during composite processing.

To optimise magnetic interaction with the composite material while ensuring insulation and protecting the mould setup, a 150 μm thick Teflon film was used as a barrier. This thin yet durable layer minimised the distance between the magnets and the composite, allowing sufficient magnetic exposure to orient the magnetite particles effectively. At the same time, the Teflon film preserved the mechanical and thermal integrity of the setup during the heating and compression stages.

As illustrated in Figure 2, the consistent magnetic flux density of 350 mT, combined with the precise positioning of the magnets in an NS configuration, created a stable magnetic field gradient between 0.2 mm and 3 mm from the magnet surface. Both the flux density and field direction were carefully controlled to ensure reproducibility of the process and its results. Additionally, exposure duration and process temperature were varied to investigate their effects on particle alignment within the composite.

### 2.3. Composite Characterisation

#### 2.3.1. Morphological Analysis

Micro CT analysis: The micro-CT assessed composites’ three-dimensional (3D) microstructural and compositional heterogeneities. Digital radiographs were acquired on a micro CT SkyScan 1275 scanner (Bruker, Billerica, MA, USA) using an X-ray cone incident on a rotating specimen. The following operating conditions were used: source voltage of 48 kV, current of 80 µA, a distance of 9–10 µm and an average of four radiographs per position. The acquisition was performed by rotating the specimen over 360° with a 0.25° rotational step. The reconstructions were obtained with the NRecon (version: 2.2.0.6), and volumetric visualisation was achieved with CTvox program (3.3.1), which integrate the instrument software packages.

SEM analysis: The MWCNT and the magnetite on the cross-section of the polymeric tapes was visualised using a Field Emission Gun Scanning Electron Microscope (FEG-SEM, NOVA 200 Nano SEM, FEI Company, Hillsboro, OR, USA). Before analysis, specimens were cryo-fractured using liquid nitrogen and coated with a 1.5 nm film of Au-Pd (20–80 weight %) using a high-resolution sputter coater (208HR, Cressington Company, Watford, UK), coupled with an MTM-20 High Resolution Thickness Controller (Cressington). The topographic images were obtained with a secondary electron (SE) detector at an acceleration voltage of 10 kV. The atomic contrast images were obtained with a Backscattering Electron Detector (BSED) at an acceleration voltage of 15 kV. The chemical analyses of samples were performed with the Energy Dispersive Spectroscopy (EDS) technique, using an EDAX Si(Li) detector at an acceleration voltage of 20 kV.

#### 2.3.2. Rheological Analysis

Oscillatory shear rheological analysis was conducted to evaluate the impact of MWCNT and magnetite on the composites’ viscosity, loss, and storage modulus. The experiments were performed using a Discovery HR 10 rheometer from TA Instruments (New Castle, DE, USA), which provides precise control and measurement capabilities. The tests were conducted at various temperatures (160, 190, and 220 °C) to assess the thermal influence on rheological properties. The analysis employed a parallel stainless-steel plate geometry ideal for uniform shear distribution. The linear viscoelastic region was identified through a stress sweep at a frequency of 1 Hz to ensure accurate characterisation of the materials’ viscoelastic behaviour. Initially, a frequency sweep analysis was conducted on all samples ranging from 0.1 to 1000 rad/s. This test provided insights into how the composites respond to dynamic shear stresses over a wide frequency range. Subsequently, a time sweep analysis, at a frequency of 1 Hz, was performed during 20 min within the linear viscoelastic region. This analysis focused on samples containing 1 wt.% MWCNT, 6 wt.% magnetite, and the combined 1 wt.% MWCNT with 6 wt.% magnetite. This allowed for the assessment of time-dependent changes in rheological properties, offering a detailed understanding of the stability and performance of the composites under sustained shear conditions at temperatures of 160, 190 and 220 °C.

#### 2.3.3. Thermogravimetric Analysis (TGA)

TGA was conducted using the equipment TGA 209 F1 Libra instrument from Netzsch (Bavaria, Germany) to evaluate the thermal stability and thermal decomposition of the composites produced. The TGA was performed under a nitrogen atmosphere. Each sample was heated from 30 to 700 °C at a consistent rate of 20 °C/min. Alumina crucibles were used throughout the testing process. The samples tested consisted of composites with 1 wt.% MWCNT, 3 wt.% MWCNT, 1 wt.%, 3 wt.% and 6 wt.% magnetite, and a combination of 1 wt.% MWCNT with 6 wt.% magnetite and 3 wt.% MWCNT with 6 wt.% magnetite. These concentrations were selected to investigate the influence of MWCNT and magnetite loadings on the thermal properties of the composites. To ensure that the observed changes in rheological behaviour over time were not due to thermal degradation, TGA was conducted over a temperature range from ambient to 220 °C, and maintained at 220 °C for 20 min.

#### 2.3.4. Electrical and Functional Testing

Electrical resistivity analysis: Electrical resistivity measurements for the composites were performed using the two-point electrical resistance method, employing a Keithley 6487 picoammeter/voltage source. To ensure electrical contact, the cross-sections of the opposite sides of each composite sample were coated with silver ink (CI 1036, Engineered Materials Systems Inc., Delaware, OH, USA). The dimensions of each specimen were 40 mm in length, 20 mm in width, and 3 mm in thickness, yielding an electrode cross-sectional area of approximately 60 mm^2^. After applying silver ink, the composites were thermally treated at 110 °C for 10 min to ensure the cure of the ink, its adhesion and conductivity. Electrical resistance measurements were conducted on five replicates per composite formulation.

Sensor fabrication: To explore a complementary application of MWCNT/magnetite/TPE composites in flexible electronics, laser-cut patterns with 1 mm wide zig-zag lines were fabricated using a Laser Spirit LS machine (GCC, New Taipei City, Taiwan). These patterns were then bonded to a polyimide (PI) substrate pre-printed with silver ink electrical pads.

Sensor pattern positioning was achieved using SmCo permanent magnets (characterised in Figure 2) with a surface magnetic flux density of 350 mT. Arranged in a north–south configuration (Figure 2), these magnets generated a strong, uniform magnetic field, even at elevated temperatures. This field facilitated precise alignment and secure positioning of the sensor patterns, allowing adjustment of the zig-zag line spacing for specific applications. The magnetic force ensured uniform pressure across the sensor patterns, enhancing adhesion to the PI substrate during the subsequent 220 °C, 10 min bonding process in an oven. This temperature was carefully selected to avoid melting the substrate while enabling the TPE to melt and form a strong bond between the composite and the substrate. The pressure required for bonding was provided by the magnetic interaction between the SmCo magnets and the magnetite particles within the composite. After the heating process, the magnets and sensor were carefully removed from the oven and allowed to cool to ambient temperature to stabilise the bonded structure.

The electrical resistance of the bonded sensor patterns was measured using a Keysight U1282A multimeter (Keysight Technologies, Santa Clara, CA, USA) connected to a computer, enabling precise data acquisition to verify sensor functionality and adhesion quality to the PI substrate. This magnetic field-assisted fabrication process simplifies sensor positioning and ensures robust adhesion and alignment, offering a practical and scalable solution for flexible electronics.

Sensor stability testing: The preliminary analysis of the sensor’s stability was assessed by performing 100 bending cycles under 10 mm deformation in a Shimadzu AGX-V (Shimadzu Corporation, Kyoto, Japan) at the rate of 150 mm/min. The stability of the sensor was also tested with 10 mm deformation and 10 s holding under deformation cycles. Then, an extra bending step with 5 mm deformation was tested, followed by a 10 mm deformation step.

## 3. Results and Discussion

### 3.1. Morphological Analysis

To investigate the impact of the magnetic field on particle orientation, a 6 wt.% magnetite/TPE composite was analysed using micro-CT imaging. The micro-CT images, shown in Figure 2c,f, reveal variations in particle density within the TPE matrix. Brighter areas in these images indicate higher concentrations of magnetite, as magnetite absorbs more X-rays due to its higher density. The micro-CT analysis provides insight into particle orientation patterns induced by the magnetic field. In addition to the observed particle orientation with the magnetic field lines, the images reveal a hole in the composite, which aligns with regions of low magnetic field intensity from the magnetic mapping data. This phenomenon suggests that magnetite particles are drawn toward areas of higher magnetic flux density, as illustrated in the magnetic mapping images in Figure 2b,e. The movement of magnetite particles toward stronger magnetic fields creates defects in regions with weaker magnetic fields. The formation of such defects could be reduced in composites with a low-viscosity polymer matrix, as shown in studies performed on thermoset polymers like PDMS and PU [11,31]. On the one hand, in low-viscosity thermoset matrices, magnetic particles can move more freely toward areas of higher magnetic flux density, allowing particles to accumulate in regions of strong magnetic fields without dragging the surrounding matrix. On the other hand, high-viscosity thermoplastic matrices like TPE behave differently. Here, the viscosity is high enough that magnetic particles cannot move independently. Instead, as the particles attempt to migrate toward stronger magnetic fields, they also drag the surrounding polymer matrix, creating defects in the composites. The magnetic field mapping highlighted the influence of the magnetic poles on particle orientation within the composite. The presence of north and south poles affects how magnetite particles orient and aggregate, impacting the material’s density, as observed in the micro-CT images (Figure 2c,f). The images reveal that the magnetite particles align parallel to the north–south poles direction. With the four magnets arranged in a north–south configuration, the particles exhibit a circular orientation around the central region, with low magnetic flux density.

The particle orientation mechanism was studied through a detailed analysis of the effect of the composition and processing parameters on the orientation of magnetite. In particular, the effects of magnetite concentration and compression moulding parameters were analysed using micro-CT imaging to visualise changes in particle orientation, as shown in the cross-section images in Figure 3. The impact of the weight percentage of magnetite on the particle alignment is depicted in Figure 3a, showing that 1 wt.% of magnetite is insufficient to induce a uniform magnetic alignment. However, increasing magnetite content to 3 and 6 wt.% leads to particle orientation in the magnetic field, as shown in Figure 3b,c. At lower magnetite concentrations the distance between particles is too large to facilitate their interaction in the magnetic field. This leads to weaker magnetic interactions within the material, hampering the attainment of uniform magnetic alignment. Additionally, the magnetic moments of individual particles are often insufficient to counteract the random thermal motion and orientation within the polymer matrix. In contrast, at higher concentrations (3 and 6 wt.%), the density of magnetic particles in the composite is large enough to induce an overall magnetic responsiveness [39]. The shorter distance between particles fosters stronger collective magnetic interactions, facilitating more effective alignment in response to a magnetic field. Furthermore, the increased particle density provides a continuous and interconnected pathway for magnetic forces to act upon, thereby improving alignment uniformity [40,41].

Additionally, the micro-CT analysis of process parameters, specifically temperature and moulding time, reveals that increasing the processing time enhances the orientation of magnetic particles within the composite, Figure 3d–f. Significant particle alignment is evident within the initial 10 min, indicating that an efficient orientation process can be achieved relatively fast, thus conserving time and resources. Decreasing the compression moulding time to 5 min hampers particle orientation, as it is not enough time for the polymer to reach 190 °C. Concerning temperature effects, there is a positive correlation between elevated temperatures and improved particle alignment, Figure 3g–i. Setting a temperature of 220 °C effectively reaches a mould cavity temperature of approximately 190 °C after 10 min and improves particle alignment. This temperature setting is consistent with the recommendations provided in the polymer datasheet, ensuring that processing conditions are ideal without compromising the integrity of the material. To further confirm the magnetite orientation in the magnetic field observed by micro-CT analysis, the particle alignment was confirmed by SEM imaging and represented in Figure 4a. The brighter particles observed in the Micro-CT images were analysed using SEM/EDS, as shown in Figure 4. Back scattered electrons (BSE) mode imaging was used to identify elements with higher atomic numbers, allowing straightforward detection of magnetite.

EDS analysis then confirmed the elemental composition of the analysed areas, showing strong iron peaks in magnetite-rich regions and lower iron signals in areas with less magnetite (Figure 4b and Table 2).

In addition to the BSE analysis of composites with magnetite, the interaction between magnetite and MWCNT in the same polymer matrix was examined, as shown in Figure 5. The results showed that specimens containing 1 wt.% MWCNT and 6 wt.% magnetite, subjected to a magnetic field in the melt state, demonstrated clear alignment along the magnetic field lines near the location of the permanent magnet, where the magnetic flux density is highest (Figure 5b). This structured orientation is notably less pronounced when compared to the pure 6 wt.% magnetite sample (Figure 4a), suggesting that the presence of MWCNT affects the particle orientation process, possibly by increasing the composite viscosity, which was confirmed by the rheological results. The composite with 3 wt.% MWCNT and 6 wt.% magnetite depicts even lower orientation of the magnetite particles (Figure 5c), resulting from the higher viscosity of this composite with higher MWCNT concentration. The resulting pattern of magnetic particles resembles that of the composite with 1 wt.% MWCNT and 6 wt.% magnetite fabricated in the absence of a magnetic field (Figure 5a).

Figure 6 shows the morphology of the 1 wt.% MWCNT and 6 wt.% magnetite composite prepared with the application of a magnetic field, at higher magnification, to investigate the interaction between magnetite and MWCNT further. In these images, the alignment of magnetite and MWCNT particles in response to the magnetic field is visible, with BSE imaging providing evident contrast (Figure 6a,c). The images effectively highlight the alignment and distribution of magnetite particles along the magnetic field lines, with MWCNTs also showing some orientation induced by the localised movement of magnetite within the matrix. In the absence of a magnetic field, magnetite particles and MWCNTs are dispersed randomly throughout the TPE matrix (Appendix A).

### 3.2. Rheological Analysis

The rheological analysis of TPE composites containing varying concentrations of MWCNT and magnetite provides insight into the influence of these fillers on the properties of the composite melt. The results indicate that MWCNTs have a significant impact on the complex viscosity, loss modulus (G”), and storage modulus (G’) of the composites. In contrast, adding magnetite at 1, 3, and 6 wt.%, shows minimal effect on complex viscosity, loss modulus, or storage modulus, compared to the pure TPE matrix (Figure 7a–d). The similarity in the rheological response of TPE and magnetite-filled TPE composites, and the large shift observed for the nano-size MWCNT/TPE composites, confirms the influence of the particle size on the melt rheology. However, the combination of MWCNT with magnetite leads to a slight increase in complex viscosity, storage modulus, and loss modulus compared to the samples containing only MWCNT. This synergistic interaction between MWCNT and magnetite enhances the material’s storage modulus and its ability to dissipate energy. Furthermore, the analysis shows that the complex viscosity of all materials decreases with increasing frequency, which suggests that the alignment of polymer chains under shear may contribute to a reduction in viscosity. This shear-thinning behaviour is characteristic of viscoelastic materials and highlights the importance of polymer chain dynamics in determining the flow properties of the composite [42].

Interestingly, it is observed that the composite containing 6 wt.% magnetite presents the loss modulus consistently above the storage modulus, across the tested frequency range (Figure 7b). This indicates the dominant viscous component of the complex modulus (G” > G’), with energy dissipation dominating over elastic energy storage. In contrast, the samples containing 1% MWCNT and those containing 1% MWCNT combined with 6% magnetite exhibit a frequency-dependent transition: the viscous component dominates at low frequencies (G” > G’), however, as the frequency increases, the storage modulus becomes dominant (G’ > G”). This transition suggests that at higher deformation rates, the material’s ability to store energy elastically becomes more significant, possibly due to the enhanced network formation and interaction between the MWCNTs and the polymer matrix.

A time sweep analysis was performed at different temperatures to further investigate the rheological behaviour of the composites. Figure 8 illustrates individual materials storage and loss modulus at 160, 190, and 220 °C. This analysis depicts the evolution of the material’s viscoelastic properties over 20 min at constant temperature. Additionally, a comparison of the complex viscosity for all samples was conducted at 160 and 190 °C, reflecting the highest real temperature achieved during the compression moulding process, which was 190 °C. The storage and loss modulus analysis extended to 220 °C to study the influence of higher temperature on the polymer’s rheological behaviour.

The rheological studies revealed the influence of fillers such as MWCNT and magnetite, as well as their combination, on the elastic and viscous responses of the composite materials under constant thermal conditions. For the TPE composites, illustrated in Figure 8a, storage and loss modulus remain relatively stable over time at temperatures of 160 and 190 °C, indicating stable elastic and viscous properties within this range. However, at 220 °C, the storage modulus shows a noticeable increase, suggesting that the material undergoes structural changes, possibly due to the onset of cross-linking or other thermal effects, which enhance its elastic response [43]. The loss modulus for TPE does not exhibit significant changes, implying that its ability to dissipate energy as heat remains constant across the temperature range.

The TPE composite containing 6 wt.% magnetite (Figure 8b) depicts a pronounced increase in the storage modulus at both 190 °C and 220 °C, exhibiting a response similar to TPE but starting earlier and reaching a greater magnitude. This suggests that magnetite significantly enhances the material’s stiffness as temperature increases, making the composite more resistant to deformation under elastic strain. At 160 °C, the loss modulus remains higher than the storage modulus over the 20 min, indicating a predominantly viscous behaviour. However, as the temperature rises to 190 °C, the storage modulus approaches or exceeds the loss modulus, signalling a shift towards an elastic behaviour, with the material achieving gelation in less than 15 min. At 220 °C, gelation occurs faster, within 7 min, indicating that at this temperature the initial storage modulus is lower than at 190 °C but quickly surpasses the storage modulus observed at 160 °C. This demonstrates that the composite rapidly transitions to a more elastic state at elevated temperatures, highlighting the significant impact of temperature on its rheological properties.

The TPE composite containing 1 wt.% MWCNT (Figure 8c) shows the storage and loss modulus stable behaviour at 160 and 190 °C, with a pronounced increase at 220 °C, particularly in the storage modulus. This suggests that MWCNTs contribute to the material’s energy dissipation capabilities, enabling gelation in approximately 7 min at 220 °C, although not to the extent observed in the sample with 6 wt.% magnetite. This observation is further supported by the analysis of the composite containing 1 wt.% MWCNT combined with 6 wt.% magnetite, which shows the most significant changes across the studied temperatures (Figure 8d). At 160 °C, the storage modulus displays a balanced increase, reflecting the synergistic effect of MWCNT and magnetite in enhancing the composite’s elastic properties over time. At 190 °C, the storage modulus increases substantially, indicating that the combination of these fillers strongly reinforces the material’s elastic network at higher temperatures. At 220 °C, the test begins with G’ already higher than G”, indicating that the composite is stiffer at this temperature compared to other tested temperatures, highlighting the significant reinforcement provided by both MWCNT and magnetite.

Finally, the evolution of complex viscosity of the three composites at 160 °C and 190 °C over 20 min is compared in Figure 8e. At 160 °C, complex viscosity remains relatively stable throughout the 20 min test period. However, at 190 °C, after some time a noticeable increase in viscosity over time is observed, particularly for composites containing 6 wt.% magnetite, and 1 wt.% MWCNT with 6 wt.% magnetite. To provide insight into the contradictory enhanced particle alignment observed in Figure 3f and Figure 8f show the complex viscosity of the composites under a range of temperatures, simulating the mould cavity temperature evolution. With a set temperature of 220 °C, the cavity temperature gradually increases from 130 to 190 °C, allowing the sample to remain at lower temperatures initially. This gradual heating keeps the viscosity lower than at constant temperatures of 190 °C, delaying crosslinking that would otherwise increase viscosity. As a result, particle mobility and alignment are enhanced within the composite material. This gradual temperature increase reflects the real moulding process, where sustained lower viscosity over 20 min facilitates alignment.

### 3.3. Thermogravimetric Analysis (TGA)

Figure 9 presents the results of the thermogravimetric analysis (TGA). The TGA curves in Figure 9a show that incorporating MWCNTs and magnetite at the specified concentrations significantly enhances the thermal stability of the composites compared to the neat TPE matrix. The thermal decomposition of the composites may be characterised by two parameters: the temperature at which mass losses of 5% (T5%) and 50% (T50%) are reached. As detailed in Table 3, composites containing 1 wt.% MWCNT and either 1 wt.% or 3 wt.% magnetite exhibit a T5% comparable to that of the neat TPE. However, composites with 3 wt.% MWCNT show a notable 20 °C increase in T5%, while those with 6 wt.% magnetite exhibit a 12 °C increase. This behaviour aligns with previous studies [44,45,46], where the thermal stability increase was attributed to the heat dissipation properties of MWCNTs and magnetite and their ability to hinder the diffusion of volatile degradation products. Notably, the synergistic interaction between MWCNTs and magnetite amplifies this stabilising effect. The composite containing 1 wt.% MWCNT with 6 wt.% magnetite exhibits a 16 °C increase in T5%, while the composite containing 3 wt.% MWCNT with 6 wt.% magnetite shows a more pronounced increase of 36 °C. A similar trend is observed for T50%, the fillers delaying thermal decomposition for all composite samples. These results confirm the potential of MWCNT and magnetite to increase the thermal stability of TPE-based composites.

In a separate test (Figure 9b), the thermal stability of a composite containing 6 wt.% magnetite was assessed by applying a controlled temperature ramp from ambient temperature to 220 °C over a period of 10 min, and then holding for 20 min at that temperature. During the isothermal stage, the weight loss percentage (TG%) remained nearly constant, with a small variation of approximately 1% observed, possibly due to residual humidity within the composite, showing the thermal stability at this temperature.

### 3.4. Electrical and Functional Testing

The electrical resistivity characterisation of the composites, formulated with different concentrations of MWCNT and magnetite within the TPE matrix, was done, revealing the influence of magnetite on the composite’s resistivity (Table 4). Specifically, samples containing magnetite without a magnetic field applied during production exhibited slightly higher resistivity than those with an applied magnetic field. This suggests that simply adding magnetite without subsequent applied magnetic field may not enhance conductivity and might slightly disrupt the MWCNT conductive network. Conversely, applying a magnetic field during composite production appears to positively impact resistivity. This effect was particularly evident in the 1 wt.% MWCNT composites with 6 wt.% magnetite, where samples prepared under a magnetic field exhibited an electrical resistivity of (8.80 ± 7.51) × 10^1^ Ω.m compared to those processed without magnetic alignment, with (4.55 ± 6.94) × 10^5^ Ω.m, or the control sample with just 1 wt.% MWCNT and electrical resistivity of (1.43 ± 2.76) × 10^5^ Ω.m. Increasing the MWCNT content to 3 wt.% further decreased resistivity, especially when combined with an applied magnetic field on the mould cavity, achieving a value of (9.52 ± 6.11) × 10^−1^ Ω.m. These preliminary results indicate that including magnetite on conductive composites produced under a magnetic field seems to decrease their electrical resistivity. This increase could be attributed to the orientation of particles, which is reported in the literature as an effective method for enhancing electrical conductivity [47], or potentially due to some degree of internal particle compaction. 

The variability in electrical resistivity, particularly for composites with low MWCNT concentration near the percolation threshold, is frequently observed. Small variations in nanoparticle dispersion and distribution across the composite can significantly impact conductivity [48]. To mitigate this variability, more precise feeding systems may be used to improve the control on MWCNT concentration during extrusion. However, this is a general limitation for compositions of small additive concentrations. Alternatively, a masterbatch dilution approach may be adopted, pre-dispersing a high concentration of MWCNT in the polymer matrix and then diluting it to the desired lower concentration. This method enhances the nanoparticle distribution and reduces challenges associated with feeding low-density materials. 

Despite the observed variability in electrical resistance, the results highlight the potential of using magnetic fields and magnetite to align and distribute conductive fillers within the matrix. This approach is particularly advantageous for applications involving low-viscosity polymers and processing techniques like compression or injection moulding, as it effectively decreases electrical resistivity and enhances composite conductivity. Another potential application of magnetite in conductive composites involves sensor positioning and geometry control. This involves producing films from these composites, laser cutting sensor patterns, and then using permanent magnets to precisely immobilise the sensor pattern on flexible substrates, as demonstrated in Figure 10. This magnetic positioning technique applies pressure to the sensor pattern, allowing it to bond to the flexible substrate under temperature. The elevated curing temperature of 220 °C ensures adhesion between the sensor and the PI substrate, providing resistance to delamination even under cyclic deformation, as shown in Figure 10e,f [49].

To demonstrate the stability and functionality of the sensor, the composite with 3 wt.% MWCNT and 6 wt.% magnetite was assembled into a bending sensor and subjected to 100 bending cycles. As illustrated in Figure 11a, the sensor maintained its structural integrity with no signs of delamination, which would be evident in changes of the electrical resistance values. Subsequently, the sensor underwent 10 deformation cycles with a 10 s hold at the extreme positions, Figure 11b. Then, an additional 10 cycles were performed across three distinct bending positions, as demonstrated in Figure 11c. The sensor exhibited excellent stability in detecting different levels of deformation, highlighting its value as a tool for qualitative deformation analysis. These preliminary results present a promising approach for developing resilient sensors that endure repeated deformation cycles.

Table 5 compares bending sensors produced by different methods, their general characteristics and performance, for sensors reported in the literature [50,51], to compare with the sensor developed in the present work and a commercial sensor [52]. The sensor presented in the present work has a simple fabrication process and is versatile in terms of applications. While techniques such as scraping-coating or stencil printing require specific conditions and material properties [50,51], the method proposed in this study is simple, robust and scalable. A key advantage is the use of a thermoplastic elastomer (TPE) instead of thermoset polymers as PDMS, commonly used in similar studies [50,51]. TPE advantages include recyclability, high processability, and compatibility with screen printing, simplifying the often challenging printing of electrodes and contacts in PDMS-based bending sensors. TPE is cost-effective, its composites may be produced using large-scale production methods, making it highly suitable for industrial applications. These advantages position the sensor proposed in this work as a promising candidate for applications in wearable electronics, robotics, and soft robotics. Furthermore, integrating advanced techniques, such as laser cutting and incorporating magnetite and MWCNT composites, aligns these sensors with contemporary advancements in bio-inspired designs and graphene-based devices, joining multifunctionality and durability under mechanical stress [53,54]. The robustness and operational stability demonstrated through repeated deformation cycles, combined with the adaptable design, broadens the scope of potential applications in flexible electronics and multifunctional actuator systems [55].

## 4. Conclusions

This work highlights the versatility of magnetite as an additive in flexible and electrically conductive composites, emphasising its promise in magnetic field-assisted orientation and positioning applications. Magnetite’s magnetic responsiveness enables precise particle orientation within the thermoplastic elastomer (TPE) matrix, significantly enhancing the electrical conductivity of MWCNT/magnetite composites when exposed to a magnetic field. Magnetite does not impact TPE viscosity, facilitating particle mobility and alignment under magnetic influence. This characteristic enables the distribution of MWCNT particles when combined with 6 wt.% magnetite, enhancing the formation of conductive pathways and reducing electrical resistivity. This demonstrates magnetite’s suitability in applications requiring controlled particle orientation to boost conductivity without compromising the processability of the composite. Additionally, preliminary results were obtained for a bending sensor composed of 3 wt.% MWCNT and 6 wt.% magnetite, precisely adjusted and immobilised on a flexible substrate using permanent magnets, demonstrated stable performance across multiple bending positions and cycles, indicating a promising approach for flexible electronics.

Future work should refine these findings by exploring additional or alternative conductive additives to optimise composite performance. Improved MWCNT feeding methods to produce composites with low nanoparticle content are also relevant, to enhance particle homogeneity and reduce variability in conductivity measurements.

## Figures and Tables

**Figure 1 micromachines-16-00068-f001:**
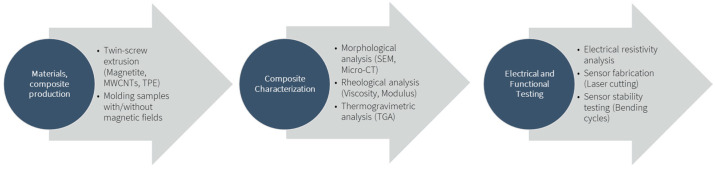
Experimental Workflow.

**Figure 2 micromachines-16-00068-f002:**
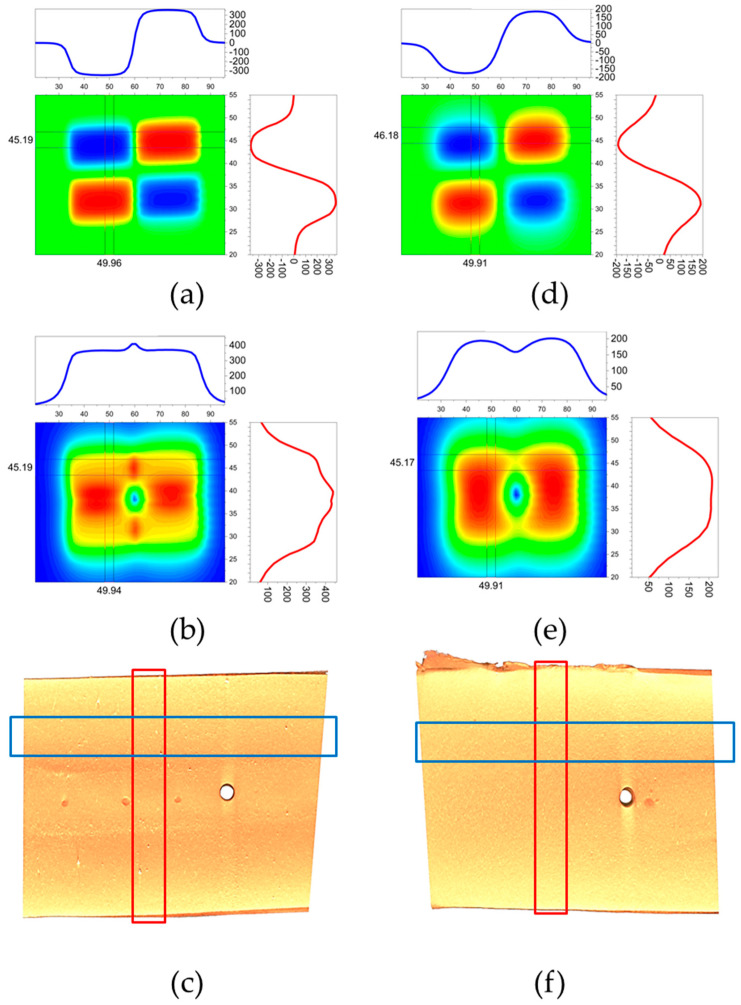
Magnetic field mapping and micro-CT analysis of a 6 wt.% magnetite composite. The magnetic flux density (mT) is represented using a colour scale, with green to red indicating increasing intensity. The blue line at the top of each subplot represents the magnetic flux density in the x-direction (horizontal), while the red line on the right side of each subplot shows the magnetic flux density in the y-direction (vertical). (**a**) NS polarity at a distance of 0.2 mm (**b**) magnetic field interaction between north and south pole at 0.2 mm (**c**) and bottom view of the sample, near the location of the permanent magnets. (**d**) NS polarity at a distance of 3 mm (**e**) magnetic field interaction between north and south pole at 3 mm (**f**) and top view of the sample, 3 mm away from the location of the permanent magnets.

**Figure 3 micromachines-16-00068-f003:**
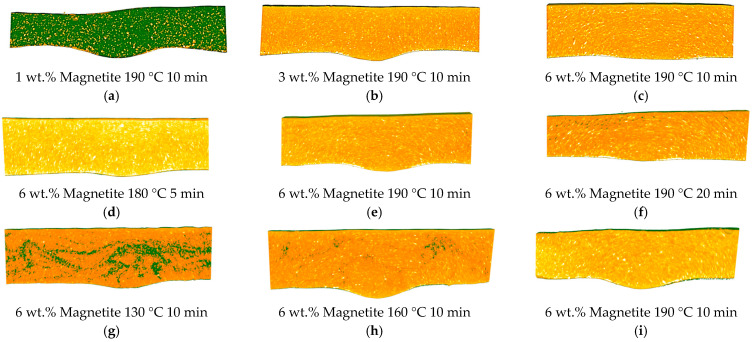
Micro-CT images of cross-section samples: (**a**) 1% magnetite with 220 °C set temperature, during 10 min; (**b**) 3% magnetite with 220 °C set temperature, during 10 min; (**c**) 6% magnetite with 220 °C set temperature, during 10 min; (**d**) 6% magnetite with 220 °C set temperature, during 5 min; (**e**) 6% magnetite with 220 °C set temperature, during 10 min; (**f**) 6% magnetite with 220 °C set temperature, during 20 min; (**g**) 6% magnetite with 160 °C set temperature, during 10 min; (**h**) 6% magnetite with 190 °C set temperature, during 10 min; (**i**) 6% magnetite with 220 °C set temperature, during 10 min.

**Figure 4 micromachines-16-00068-f004:**
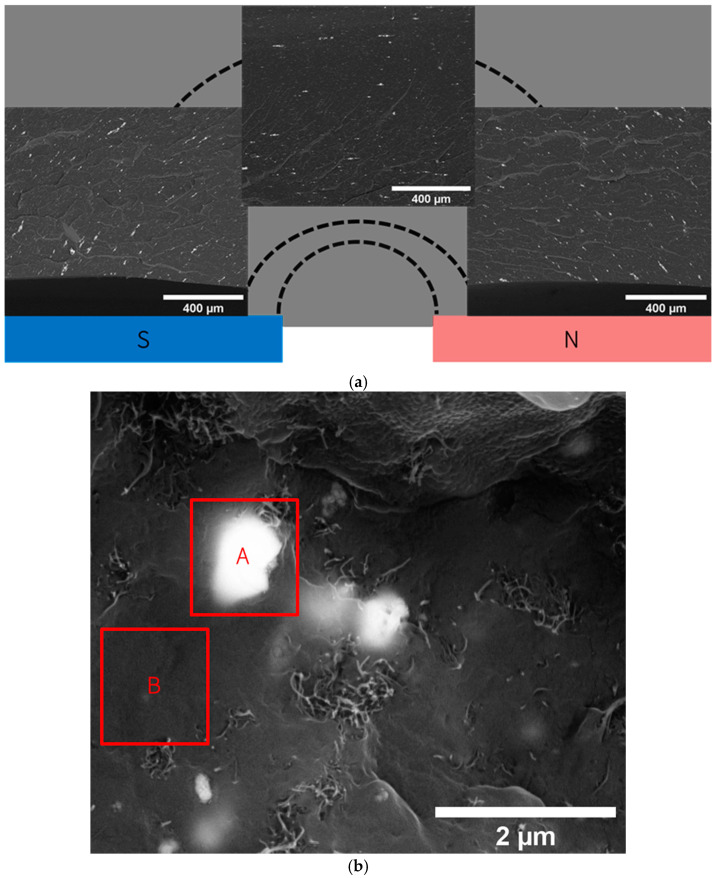
SEM images: (**a**) Reconstruction of the magnetite alignment inside the TPE composite after being fabricated under a static magnetic field. (**b**) with BSED analysis to demonstrate the correspondence of the white spots (A) to the location of magnetite, and black region (B), to the location of MWCNT and polymer.

**Figure 5 micromachines-16-00068-f005:**
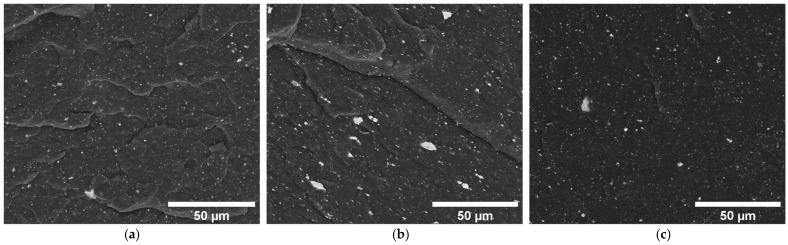
SEM images of the specimen cross-section at the corresponding magnifications performed in the same region: (**a**) 1% MWCNT_6% Magnetite without an applied magnetic field, (**b**) 1% MWCNT_6% Magnetite with an applied magnetic field, and (**c**) 3% MWCNT_6% Magnetite with an applied magnetic field.

**Figure 6 micromachines-16-00068-f006:**
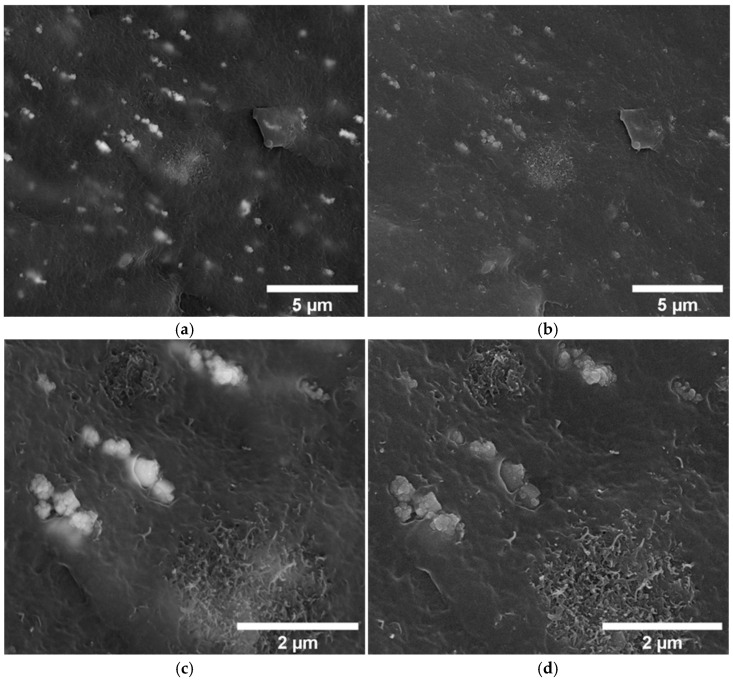
SEM images of a composite with 1% MWCNT and 6% magnetite under an applied magnetic field, shown in: (**a**) BSE, (**b**) SE modes, (**c**) BSE, and (**d**) SE modes at different magnification.

**Figure 7 micromachines-16-00068-f007:**
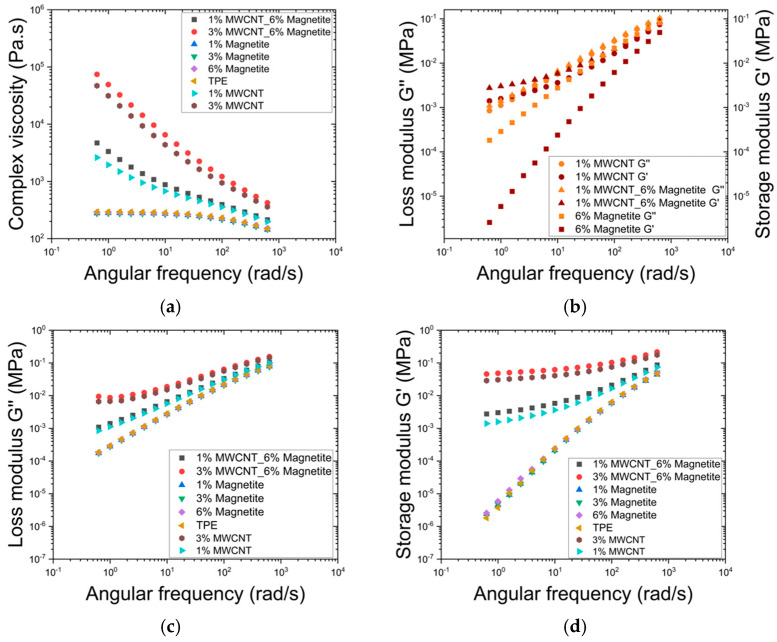
Rheology results obtained by frequency sweep for TPE, 1,3 and 6% magnetite, 1 and 3% MWCNT and the combination of 1% MWCNT with 6% magnetite and 3% MWCNT with 6% magnetite composites performed at 190 °C. (**a**) complex viscosity; (**b**) storage modulus and loss modulus for 1% MWCNT, 6% magnetite and 1% MWCNT with 6% magnetite composites; (**c**) storage modulus; and (**d**) loss modulus for all the composites.

**Figure 8 micromachines-16-00068-f008:**
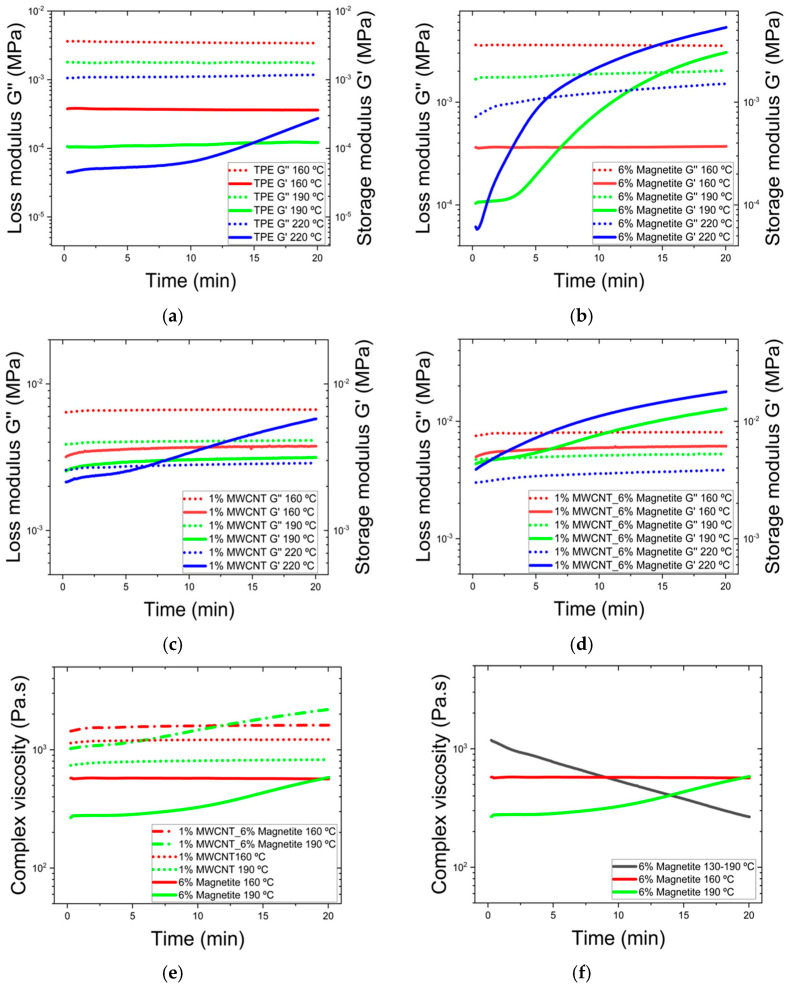
Time sweep rheology results, with storage and loss modulus for (**a**) TPE at 160, 190 and 220 °C; (**b**) 6% magnetite at 160, 190 and 220 °C; (**c**) 1% MWCNT at 160, 190 and 220 °C; (**d**) 1% MWCNT with 6% magnetite at 160, 190 and 220 °C; and (**e**) complex viscosity for the samples of 1% MWCNT, 6% magnetite and 1% MWCNT with 6% magnetite at 160 and 190 °C. (**f**) Complex viscosity of TPE composites containing 6% magnetite at various temperature conditions: constant temperatures of 160 °C and 190 °C, and a gradually increasing temperature from 130 °C to 190 °C.

**Figure 9 micromachines-16-00068-f009:**
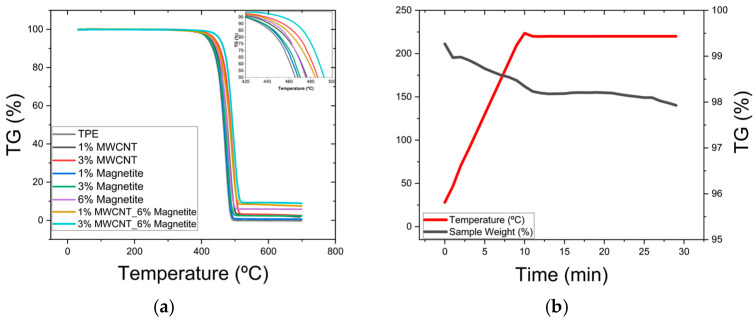
(**a**) TGA for the 1, 3 and 6 wt.% magnetite, TPE, 1% MWCNT with 6 wt.% magnetite and 3 wt.% MWCNT with 6 wt.% magnetite samples. (**b**) isothermal thermogravimetric analysis for the sample with 6% magnetite at 220 °C for 20 min.

**Figure 10 micromachines-16-00068-f010:**
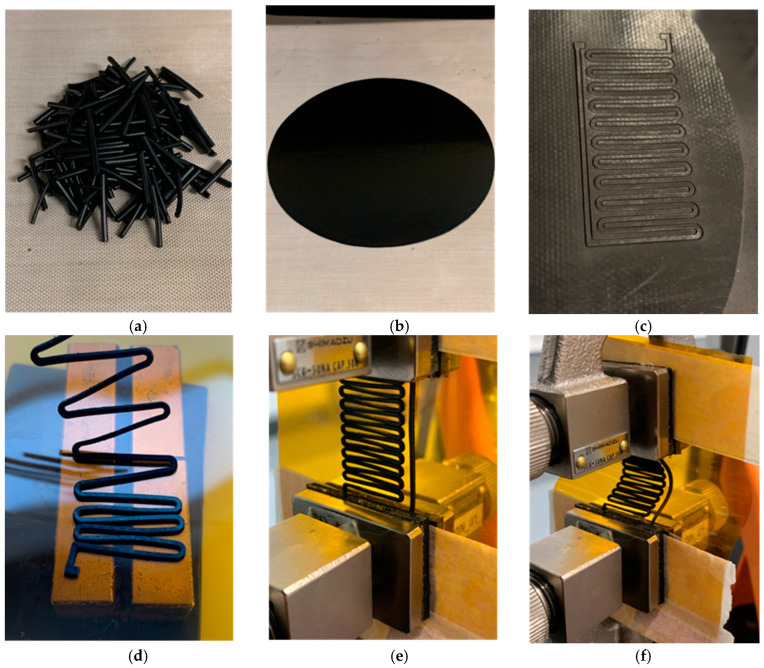
(**a**) extruded material, (**b**) compression moulding step to create a 0.3 mm film, (**c**) laser cutting process, (**d**) sensor positioning under magnetic field, (**e**) sensor bonded to the polyimide substrate, (**f**) sensor testing under cyclic bending.

**Figure 11 micromachines-16-00068-f011:**
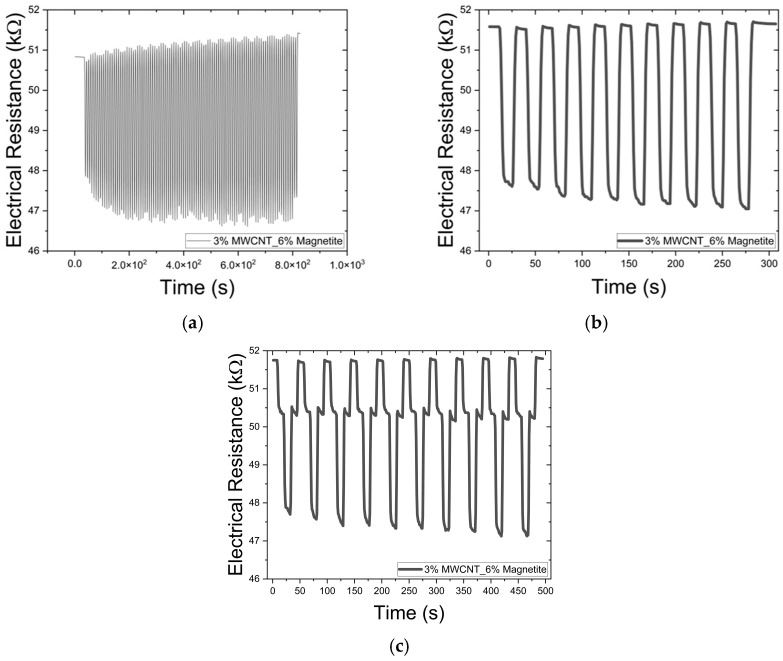
(**a**) Demonstration of the sensor stability under 100 bending cycles at 150 mm/min. (**b**) 10 bending cycles with 10 s holding time between bending cycles. (**c**) demonstrating the stability of the sensor at three different bending phases (no bending, medium bending, and full bending).

**Table 1 micromachines-16-00068-t001:** Summary of process conditions for the composites.

Sample	Magnetite Concentration (wt.%)	Set Temperature (°C)	Real Temperature (°C)	Duration (min)
a	1%	220	190	10
b	3%	220	190	10
c	6%	220	190	10
d	6%	220	180	5
e	6%	220	190	10
f	6%	220	190	20
g	6%	160	130	10
h	6%	190	160	10
i	6%	220	190	10

**Table 2 micromachines-16-00068-t002:** Normalised chemical composition of Zone A and Zone B.

Zone	Chemical Composition (At.%)		
	C	O	Fe
A	82.5	5.9	10.9
B	96.3	2.9	0.6

**Table 3 micromachines-16-00068-t003:** Thermal decomposition temperatures of the composites containing different percentages of MWCNT, magnetite and their combination.

Samples	T5% (°C)	T50% (°C)
TPE	418	465
1% MWCNT/TPE	426	476
3% MWCNT/TPE	438	486
1% Magnetite/TPE	420	468
3% Magnetite/TPE	421	470
6% Magnetite/TPE	430	475
1 wt.% MWCNT/6 wt.% Magnetite/TPE	434	484
3 wt.% MWCNT/6 wt.% Magnetite/TPE	454	492

**Table 4 micromachines-16-00068-t004:** Summary of the electrical resistivity of the composites.

Samples	Resistivity with Applied Magnetic Field (Ω.m)	Resistivity without Applied Magnetic Field (Ω.m)
1% MWCNT/TPE	-	(1.43 ± 2.76) × 10^5^
1% CNT_6%magnetite/TPE	(8.80 ± 7.51) × 10^1^	(4.55 ± 6.94) × 10^5^
3% MWCNT/TPE	-	(2.47 ± 1.62)
3% CNT_6%magnetite/TPE	(9.52 ± 6.11) × 10^−1^	(2.62 ± 2.99) × 10^1^

**Table 5 micromachines-16-00068-t005:** Comparative fabrication techniques for bending sensors.

Reference	Materials	Sensor Fabrication Method	Tested Deformation Range	Key Features	Applications
This work	3 wt.% MWCNT/6 wt.% Magnetite/TPE	Laser cutting and magnetic field assisted lamination	0–90°	Simple, scalable; good stability over 100 cycles	Wearable electronics, robotics, soft robotics
Wang et al. [50]	2 wt.% Carbon Black/PDMS	Stencil printing	−0.6% to +0.6% strain	High sensitivity; low hysteresis; scalable	Soft robotic skins, human motion detection
Zhao et al. [51]	8 wt.% MWCNT/PDMS	Scrapping coating method	0–180°	Multifunctional; good cyclic stability	Human motion/perception detection
Yu et al. [52]	Commercial bending sensor	Commercial bending sensor sewn into the fabric	180 mm curvature range	High precision; robust gesture recognition	Gesture-controlled robotics

## Data Availability

The original contributions presented in the study are included in the article/Appendix A, further inquiries can be directed to the corresponding authors.

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
