# Peer review of "Magnetic Field-Assisted Orientation and Positioning of Magnetite for Flexible and Electrically Conductive Sensors"

_micromachines, 2025, doi:10.3390/mi16010068_

Round 1

Reviewer 1 Report

Comments and Suggestions for Authors

The manuscript titled " Magnetic field-assisted orientation and positioning of magnetite for flexible and electrically conductive sensors " is of significant interest to readers in this field.  The article is well-written, well-organized, and easy to follow. I recommend its publication after addressing the following minor issues

 1.      EDS in Figure 4(b) is not clear. The authors should provide a high-resolution image to improve its clarity

2.      The authors should include XPS and TEM studies for the 1% MWCNT and 6% magnetite samples to strengthen the analysis and provide deeper insights

3.      The authors should include a comparison table showing the sensors performance relative to other studies in the literature and incorporate this table into the manuscript

4.      The authors should correct grammatical mistakes throughout the manuscript, such as 'as shown in Figure 3g), h) and i),' ensuring proper use of brackets

Author Response

Reviewer 1

  1. EDS in Figure 4(b) is not clear. The authors should provide a high-resolution image to improve its clarity.

    Dear Reviewer, thank you for your valuable feedback. To address this concern, we have replaced the EDS image with a table format (page 13) to present the data more clearly and to facilitate comparison. We believe this adjustment improves the clarity and accessibility of the information.

  1. The authors should include XPS and TEM studies for the 1% MWCNT and 6% magnetite samples to strengthen the analysis and provide deeper insights.

    We appreciate the suggestion; however, we have a limitation of equipment access and time - this work is part of a PhD thesis whose experimental part was finished and is near the final examination, and thus the time constraints prevent us from conducting additional analyses. Nonetheless, we acknowledge the importance of these characterisation techniques and plan to explore them in future studies.

  1. The authors should include a comparison table showing the sensors' performance relative to other studies in the literature and incorporate this table into the manuscript.

    Thank you for the suggestion. We have added a comparative table summarising the key parameters of our sensor alongside similar studies (Pages 23-24). This highlights our work's unique attributes, particularly the versatility and simplicity of the fabrication process. Unlike labour-intensive methods such as scraping-coating or stencil printing, our method balances simplicity, robustness, and scalability. Additionally, our use of a thermoplastic elastomer (TPE) over thermoset polymers like PDMS offers advantages such as recyclability, processability, and compatibility with screen-printing technologies, enabling cost-effective and scalable production for industrial applications.

  1. The authors should correct grammatical mistakes throughout the manuscript, such as 'as shown in Figure 3g), h) and i),' ensuring proper use of brackets.

    Thank you for pointing this out. We have reviewed the manuscript thoroughly and corrected all grammatical issues, including the improper use of brackets.

Our kind regards,

David Esteves

Reviewer 2 Report

Comments and Suggestions for Authors

1, The micro-CT and SEM images are crucial for understanding the particle orientation and distribution within the composites. However, several figures, particularly Figure 2 and Figure 4, appear to be of low resolution, making it difficult to discern details clearly. Could the authors please provide higher resolution images to ensure that all elements are clearly visible?

2, The manuscript mentions the application of a magnetic field during the composite processing. It would be beneficial to have a more detailed description of the magnetic field parameters (strength, direction, and duration) to ensure that the results can be replicated. Could the authors provide additional details on the consistency and uniformity of the magnetic field applied across different samples?

3, While the TGA results are presented, there is a lack of detailed discussion on how the addition of magnetite and MWCNTs affects the thermal stability of the composites. Could the authors elaborate on the TGA findings and how these results correlate with the composite's performance and application in high-temperature environments?

4, The electrical resistivity data presented in Table 2 shows a significant variability, especially for the samples with lower MWCNT concentrations. It would be helpful to understand the sources of this variability. Could the authors provide additional insights or conduct further analysis to explain the variability and its impact on the composite's conductivity?

5, The manuscript describes a novel method for sensor positioning using permanent magnets. Figure 10 and the associated text provide a good overview, but additional details on the process parameters (e.g., pressure, temperature, and bonding time) would be beneficial for readers to assess the practicality of this method. Could the authors include more specific details on the magnetic field-assisted sensor positioning process?

6, Some soft sensors made with low carbon materials are suggested to be introduced in the manuscript (10.1007/s10570-023-05329-y, 10.1016/j.cej.2022.139146, 10.1002/EXP.20210233)

Author Response

Reviewer 2

  1. The micro-CT and SEM images are crucial for understanding particle orientation and distribution within the composites. However, several figures, particularly Figure 2 and Figure 4, appear to be of low resolution. Could the authors please provide higher-resolution images to ensure all elements are visible?

    Thank you for your feedback. We have updated the manuscript to improve clarity and ensure that all elements are clearly visible.

  1. The manuscript mentions the application of a magnetic field during the composite processing. It would be beneficial to have a more detailed description of the magnetic field parameters (strength, direction, and duration). Could the authors provide additional details on the consistency and uniformity of the magnetic field applied across different samples?

    We appreciate your suggestion additional information was added in page 4 to clarify this aspect. Specifically, a consistent magnetic flux density of 350 mT was applied using SmCo magnets arranged in a North-South configuration, ensuring uniformity at a 3 mm distance from the magnet surface. A 150 µm Teflon film barrier was used to optimise magnetic interaction while maintaining setup integrity. These parameters, including exposure duration and process temperature, were controlled to ensure reproducibility and were varied to study their influence on particle alignment.

  1. While the TGA results are presented, there is a lack of detailed discussion on how the addition of magnetite and MWCNTs affects the thermal stability of the composites. Could the authors elaborate on the TGA findings?

    Thank you for highlighting this point. We have included a detailed discussion of the TGA results, complemented with a table comparing thermal degradation temperatures (Page 19). The data demonstrate that adding magnetite and MWCNTs significantly enhances thermal stability. This is attributed to their ability to dissipate heat and inhibit the diffusion of volatile degradation products. A synergistic interaction between MWCNTs and magnetite amplifies these effects, improving the composite's suitability for high-temperature applications.

  1. The electrical resistivity data in Table 2 shows significant variability, especially for samples with lower MWCNT concentrations. Could the authors provide additional insights into this variability?

    We have expanded on this topic in the revised manuscript (Pages 20-21). Variability is attributed to challenges in the uniform dispersion of low concentrations of MWCNTs, which can lead to irregular particle distribution. Additionally, samples with 1 wt.% MWCNT are near the electrical percolation threshold, where small variations in particle distribution can significantly impact conductivity. Strategies to mitigate this include precise feeding systems or masterbatch dilution, ensuring better dispersion and homogeneous distribution.

5    The manuscript describes a novel method for sensor positioning using permanent magnets. Could the authors include more specific details on the magnetic field-assisted sensor positioning process?

    We have added comprehensive details to the Methods section (Page 6). The sensors were positioned using SmCo magnets (350 mT) in a North-South configuration, facilitating precise alignment. During bonding, the composite was heated to 220 °C for 10 minutes, ensuring robust adhesion between the sensor and the PI substrate. Magnetic forces generated uniform pressure, enhancing adhesion. Electrical resistance measurements verified functionality, demonstrating the practicality and scalability of the process.

  1. Some soft sensors made with low-carbon materials are suggested to be introduced in the manuscript (e.g., DOI references).

    Thank you for the suggestion. We have integrated additional references, in pages 23 and 24, to enhance the manuscript's context, comparing our work to these studies.

Our kind regards,

David Esteves